# Fin Cells as a Promising Seed Cell Source for Sustainable Fish Meat Cultivation

**DOI:** 10.3390/foods14122075

**Published:** 2025-06-12

**Authors:** Zongyun Du, Jihui Lao, Yuyan Jiang, Jingyu Liu, Shili Liu, Jianbo Zheng, Fei Li, Yongyi Jia, Zhimin Gu, Jun Chen, Xiao Huang

**Affiliations:** 1Institute of Cellular and Developmental Biology, College of Life Sciences, Zhejiang University, Hangzhou 310058, China; 22207018@zju.edu.cn (Z.D.); 22307019@zju.edu.cn (J.L.); 12107043@zju.edu.cn (Y.J.); jingyu7339@gmail.com (J.L.); 2Key Laboratory of Freshwater Aquaculture Genetic and Breeding of Zhejiang Province, Zhejiang Institute of Freshwater Fisheries, Huzhou 313001, China; liushili1212@126.com (S.L.); 21207054@zju.edu.cn (J.Z.); lifeibest1022@163.com (F.L.); yongyi_jia@163.com (Y.J.); guzhimin2006@163.com (Z.G.); 3Future Food Lab, Innovation Center of Yangtze River Delta, Zhejiang University, Jiaxing 314102, China

**Keywords:** Topmouth culter, caudal fin cell, reprogramming, myogenic differentiation, adipogenic differentiation

## Abstract

Cell-cultured meat production relies on stable, proliferative seed cells, commonly sourced from muscle satellite cells (MuSCs) and adipose-derived mesenchymal stem cells (AD-MSCs). However, establishing such cell lines in fish species remains technically challenging. While pluripotent stem cells (e.g., ESCs/MSCs) offer alternatives, their differentiation efficiency and predictability are limited. Here, we developed TCCF2022, a novel caudal fin-derived cell line from Topmouth culter (*Culter alburnus*), which expresses pluripotency markers (AP, *Oct4*, *Sox2*, *Klf4*, and *Nanog*) and aggregated growth to form 3D spheroids. Forskolin supplementation enhanced pluripotency maintenance. The presence of adipogenic and myogenic lineage cells within the 3D spheroids was confirmed, demonstrating their potential as seed cells for cell-cultured meat. Using a small-molecule cocktail 5LRCF (5-Azacytidine, LY411575, RepSox, CHIR99021, and Forskolin), we successfully differentiated TCCF2022 cells into functional myotubes. Additionally, we established a method to induce the differentiation of TCCF2022 cells into adipocytes simultaneously. Thus, the TCCF2022 cell line can be used to improve muscle fiber formation and lipid composition, potentially enhancing the nutritional profile and flavor of cultured fish meat.

## 1. Introduction

With the development of society and the growth of the population, the demand for meat consumption increases continuously. However, traditional meat production industries, such as livestock, poultry, and aquaculture, have reached their limits and brought about a series of unsustainable social problems, such as the decline in wild animal resources, the loss of biodiversity, environmental pollution, and ethical and health issues [1,2,3]. Alternative protein or “artificial meat” has been considered one of the promising ways to solve these unsustainable issues. Since the first cultured beef was successfully grown in a lab [4], the emergence of cellular agriculture, which is represented by cell-cultured meat, has provided a more advantageous solution for alternative protein production in recent years [5,6,7]. Compared with other alternative protein production like plant-based proteins, cell-cultured meat is more likely acceptable to consumers duo to its utilization of cell and tissue engineering methods to create products that closely resemble natural meat in appearance, structure, nutrition, flavor, and taste [8,9].

As the largest vertebrate group, fish is an excellent model for exploring various biological processes [10] and pathophysiological mechanisms [11]. In addition, as a valuable source of high-quality protein, fish has great potential to meet the growing human demand for protein. As of now, nearly 1000 fish cell lines (URL: https://www.cellosaurus.org/, accessed on 9 June 2025) have been successfully established, and these cell lines are widely used in the research fields of cytology, genetics, immunology, ecological toxicology, etc. [12,13,14,15]. However, compared with mammals, fish cell culture technology is still relatively lagging behind, and there is little need for translational applications. Recently, the pipeline of using muscle stem cells to produce cultured fish fillets was reported for the first time in large yellow croaker [16], which sets a successful precedent for cell-cultured fish meat, but also promotes the emergence of a new field for fish cell culture translation research, i.e., “Cellular Aquaculture” [17].

The production of cell-cultured meat still faces many challenges, such as the source and expansion of seed cell lines, low-cost serum-free culture systems, large-scale expansion, and efficient directional differentiation [18,19]. Among them, the selection of seed cells plays a crucial role, and its source directly determines the formulation of subsequent operation strategies. The use of MuSCs and AD-MSCs as seed cells is favored in cultured meat biotechnology, owing to their capacity for directed differentiation into myogenic and adipogenic lineages *in vitro*. A representative application is the production of cell-cultured large yellow croaker meat, which integrates both MuSCs and AD-MSCs [16]. However, fish have very rich diversity, and the survival and expansion of cells from different fish *in vitro* requires the species-specific optimization of conditions [20]. According to our experience, it is difficult for many fish MuSCs and AD-MSCs to be subcultured *in vitro* and maintain stable proliferation and differentiation capabilities. For example, rainbow trout (*Oncorhynchus mykiss*) [21] and seabream (*Sparus aurata*) [22] muscle satellite cells are prone to spontaneous differentiation when cultured *in vitro*, making it difficult to establish stable cell lines. We have also attempted to establish MuSC lines from certain other fish species, but all efforts have failed due to premature aging, rendering them unusable for production. Pluripotent stem cells (PSCs), such as ESCs and MSCs, are also considered an ideal cell source for cell-cultured meat production due to their infinite proliferation and excellent differentiation capabilities [23]. However, the current reliance of PSCs on highly specialized culture media and intricate multi-stage differentiation processes poses significant challenges for scaling up cultivated meat production [24].

Fish fin tissues have strong regenerative ability, and fin fibroblasts from fin show excellent proliferation ability *in vitro*. A study has also shown that they can be induced to differentiate into different types of cells, such as muscle cells, adipocytes, and nerve cells [25]. Therefore, fin fibroblasts also have the potential to serve as source cells for cell-cultured fish meat. Topmouth culter is an important economic fish which is widely distributed in the major water bodies of the middle and lower reaches of the Yangtze River in China. It is favored by consumers due to its large size, fast growth, tender and delicious meat, and rich nutritional value. Its artificial breeding technology is mature and has been widely promoted, forming an aquaculture industry worth billions of Yuan [26]. However, the rich intermuscular spines of *Cyprinidae* seriously affect their edible quality and economic value. Although intermuscular spines can be removed by means of gene editing technology, it may take around ten more years to establish intermuscular spine-free strains [27]. Therefore, the production of meat through cell culture may be a more ideal way to obtain spineless and delicious fish meat.

Here, for the first time, we report the establishment and characterization of a Topmouth culter caudal fin cell line, TCCF2022, and further optimize a recipe by adding a small-molecule compound to maintain the characteristics of the cell line, including its pluripotency. Subsequently, we optimize the protocols for differentiating caudal fin cells into myotubes and adipocytes, featuring more defined components and clearer mechanisms compared to the myogenic and adipogenic induction approach reported previously [25]. In general, we establish a reliable foundation for the future application of the TCCF2022 cell line in cell-cultured fish meat production.

## 2. Materials and Methods

### 2.1. Primary Culture

The primary caudal fin culture protocol was adapted from filefish [25], with modifications to optimize. Topmouth culter fish juveniles, with a length of approximately 10 cm, were purchased from the local market in Hangzhou City, Zhejiang Province. The caudal fin tissue was collected and underwent a strict disinfection process. First, the caudal fin tissue was soaked in a 1:10,000 concentration of potassium permanganate solution for 30 min and was then washed 5 times with D-Hanks buffer (GENOM, GNM14175-5, Hangzhou, China) containing 2 × PS (Macklin, P917928, Shanghai, China) to remove excess potassium permanganate. Next, the fin tissue was disinfected by soaking in 75% alcohol for 15 s and rinsed 5 times with D-Hanks buffer to remove mucus and epithelium. Finally, the tissue was sliced into 1 mm blocks and attached to a six-well plate coated with L-type polylysine solution (Sangon, E607015, Shanghai, China) by the adhesion method. To promote the attachment growth of the tissue, the culture dish was first inverted and dried for 30 min, and then the sample blocks were digested for 10 min with 0.5 mL 0.25% trypsin–EDTA digestion solution (Cienry, CR25200, Huzhou, China). After trypsin treatment, 0.5 mL DMEM/F12 (Gibco, C11330500BT, Suzhou, China) containing 1 × PS and 10% fetal bovine serum (Gibco, 10099141C, Suzhou, China) was used to terminate digestion. Finally, 2 mL of DMEM/F12 (Gibco, C11330500BT, Suzhou, China) culture medium containing 20% fetal bovine serum, 5 ng/mL bFGF (Beyotime, P5453, purity > 96%, Shanghai, China), and 1 × PS were added to each well, and the plate was placed in a 5% CO_2_ incubator at 27 °C for culture.

### 2.2. Subculture, Cryopreservation, and Recovery

When the primary cells had reached 90% confluency, the culture medium was carefully removed, and the cells were then rinsed twice with 1 mL of D-Hanks buffer. Following this, the cells were subjected to trypsin digestion in 0.5 mL 0.25% trypsin–EDTA solution. Once the digestion was complete, 0.5 mL of complete medium was added to promptly halt the digestion. The cells were then gently aspirated with a pipette and transferred to a 1.5 mL centrifuge tube, which then was subjected to centrifugation at 300× *g* for 3 min to remove the supernatant. The cells were resuspended in complete medium (DMEM/F12 medium containing 10% fetal bovine serum, 10 ng/mL bFGF, and 1 × PS) at a ratio of one to three for subculture. If the cells needed to be frozen, 1 mL of DMEM/F12 medium containing 20% FBS and 5% DMSO (GENOM, GNM10944-1, Hangzhou, China) was utilized for resuspension, and the cells were placed in a refrigerator at −80 °C for controlled cooling. Finally, the cells were carefully transferred to liquid nitrogen for long-term preservation.

For cell recovery, cryopreserved TCCF2022 cells in cryovials were thawed in a water bath at 37 °C and suspended in 5 mL of fresh DMEM/F12. After centrifugation at 300× *g* for 3 min, the cells were resuspended in 2 mL of DMEM/F12 and seeded into 6-well plates for culture at 27 °C.

### 2.3. Culture of Forskolin-Treated Cells

For Forskolin-treated cells, we used DMEM/F12 culture medium containing 10% fetal bovine serum, 10 ng/mL bFGF, 0.5μM Forskolin (Targetmol, T2939, purity 99.86%, Shanghai, China), and 1 × PS as conditioned medium, and the plate was placed in a 5% CO_2_ incubator at 27 °C for culture. And the method of subculture was the same as in Section 2.2.

### 2.4. Growth Curve Plotting

TCCF2022 cells were seeded into 96-well plates containing complete medium according to the standard of 4 × 10^3^ cells per well, and cultured in an incubator at 27 °C under 5% CO_2_. After 1 h, when the cells successfully attached to the wall, the medium was removed and DMEM/F12 medium containing 100 UL of 10% hypersensitive cell proliferation assay (CCK-8) (Abbkine, BMU106, Wuhan, China) was added for 4 h of treatment. Cell growth was assessed by measuring absorbance at 450 nm. Subsequent tests were performed every 24 h using CCK8, and the actual absorbance was obtained by subtracting the absorbance of the blank control group. Each experimental group contained three biological replicates.

### 2.5. Karyotype Analysis

This method was adapted from the protocol adopted by Yucheng, L in 1983 [28]. T25 cell culture dish was seeded with 10^6^ cells, and when they had grown to 80% confluency, a 5 mL complete medium containing 1 ug/mL colchicine (MACKLIN, C804814, purity ≧ 98%, Shanghai, China) was added. After being incubated at 27 °C for 5 h, the medium was removed, and the cells were treated with 1 mL 0.25% trypsin–EDTA digestive solution for 5 min. Subsequently, 1 mL complete medium was added to stop the digestion, and the cells were gently blown with a pipette, transferred to a 15 mL centrifuge tube, and centrifuged at 300× *g* for 3 min. The supernatant was removed, 1 mL D-Hanks was added for cleaning once, and the cells were centrifuged again to remove the supernatant. The cells were then treated with 4 mL 0.075 M KCl for 20 min for hypotonic treatment and prefixed with 1 mL Carnot fixing solution (ethanol/acetic acid = 3:1) for 5 min. After centrifugation at 500× *g* for 5 min to remove the supernatant, 1 mL Carnot fixative was added for 30 min. The suspended cells were gently blown with a straw and centrifuged at 500× *g* for 5 min to remove the supernatant. The fixation process was repeated once, and after centrifugation, 800 µL of fixative was removed. The cells were then resuspended with the remaining 200 µL fixative. They were dropped on a slide using the cold drop method, dried at 37 °C for 30 min, and stained with 5% Giemsa dye for 30 min. Excess dye was washed away, and the glass slides were dried and then sealed with water-neutral resin. The samples were observed and photographed using a 100× oil lens.

### 2.6. Alkaline Phosphatase Staining

TCCF2022 cells were seeded into 12-well plates at a ratio of 2 × 10^5^ cells per well, and cultured in an incubator containing 5% CO_2_ at 27 °C with 1 mL of complete medium. When the cells grew to 90% confluent, the culture was continued for 1, 3, 5, and 7 days. The medium was then removed, and the cells were rinsed twice with 1 mL PBS (Sangon Biotech, B548117-0500, Shanghai, China) solution and fixed at RT for 30 min with 0.5 mL 4% PFA. After removing 4% PFA, the cells were rinsed three times with 1 mL PBS solution and were then stained with an alkaline phosphatase staining kit (red) (Applygen, E1041-50, Beijing, China) for 1 h. After the dye was removed, the dye was rinsed three times with 1 mL PBS solution and finally photographed with an inverted phase contrast microscope.

### 2.7. RT-PCR

TCCF2022 cells were seeded into a 6-well plate at a ratio of 5 × 10^5^ cells per well. After 3 days of cultivation, the medium was removed, and the cells were rinsed once with 1 mL PBS. Then, 1 mL of RNAiso plus (Takara, D9108A, Dalian, China) was added to the lysate cells and they were allowed to stand for 10 min; RNAiso plus was collected into a 1 mL centrifuge tube, 200 µL chloroform was added and the mixture was shaken violently for 15 s. The mixture was allowed to stand at room temperature for 10 min and then centrifuged at 12,000× *g* at 4 °C for 10 min, and the supernatant was removed. Isopropyl alcohol was added in equal volume and the mixture was left at room temperature for 10 min; it was then centrifuged at 4 °C 12,000× *g* for 10 min and the supernatant was removed. The mixture was rinsed once with 75% alcohol, centrifuged at 4 °C 12,000× *g* for 5 min, and dried for 5 min after removing the alcohol, and the RNA was dissolved with DEPC water. The RNA was reverse-transcribed into first Strand cDNA using a PrimeScript™ II 1st Strand cDNA Synthesis Kit (TaKaRa, 6210A, Dalian, China). The amplification of the target fragment was performed using PrimeSTAR^®^ HS DNA Polymerase (TaKaRa, DR044A, Dalian, China). See Appendix A for the RT-PCR primers.

### 2.8. qRT-PCR

Total RNA extraction and cDNA synthesis were the same as the above RT-PCR procedure. CFX ConnectTM Optics Module (BIO-RAD) and TB green (TaKaRa, RR420A, Dalian, China) were used for qRT-PCR, with β-actin as the internal reference. Each gene group was analyzed with three biological replicates, and the relative expression levels were calculated by 2^−△△CT^. See Appendix A for the qRT-PCR primers.

### 2.9. RNA Sequencing and Transcriptome Analysis Sequencing

RNA HEanalysis protocols refer to our previous work [16]. RNA libraries were constructed from the collected RNA samples utilizing the Illumina TruSeq™ RNA Library Preparation Kit and subjected to paired-end sequencing on the Illumina NovaSeq 6000 platform. Three biological replicates were analyzed per experimental group. Raw sequencing data underwent quality control processing using Fastp (v0.19.5), resulting in an average yield of ~50.5 million (SD ± 3.66 million) high-quality reads per biological replicate. De novo transcriptome assembly was conducted using Trinity software (v2.8.5). The functional annotation of the assembled transcripts was performed through BLAST+ (v2.9.0) alignment against multiple databases: NR, Swiss-Prot, Pfam, EggNOG, GO, and KEGG.

Transcript quantification was performed with RSEM (v1.3.3), with expression levels standardized to transcripts per million (TPM). Differential expression analysis was carried out using DESeq2 (v1.24.0), applying significance thresholds of |log2(fold change)| > 1 and adjusted *p*-value < 0.05, with results visualized through volcano plots. A heatmap depicting z-score normalized expression patterns of all differentially expressed genes between control (CK) and treated (F) groups was generated, implementing hierarchical clustering through the average linkage method with the Euclidean distance metric for both genes and samples.

Functional enrichment analysis was conducted through GO term classification using GOATools (v0.6.5) and KEGG pathway analysis with KOBAS (v2.1.1) for significantly differentially expressed genes.

### 2.10. Cryosection and HE Staining

The HE staining protocol refers to our previous work [18]. The 3D spheroids were collected and fixed at 4 °C overnight using 4%PFA and 30% sucrose, and then they were encapsulated using O.C.T. Compound (Scigen, 4586, Guangzhou, China). A MICROM HM525 freezing microslicer was used to slice the sections at a shelf temperature of −20 °C, and the slice thickness was 10 μm. Then, the frozen slices were placed in distilled water for 2 min, stained with hematoxylin staining solution for 5 min, soaked in tap water to rinse off the excess staining solution for 10 min, washed again with distilled water, dehydrated with 95% ethanol for 5 s, dyed with eosin staining solution for 30 s, dehydrated with 95% ethanol for 2 min, and then dehydrated with fresh 95% ethanol for 2 min. Xylene transparent was added for 5 min and then replaced with fresh xylene transparent for 5 min, before being sealed with neutral gum.

### 2.11. Immunofluorescence and Histological Staining of Cryosections

Immunofluorescence and Histological staining protocols refer to our previous work [18]. Immunostaining was performed by a general protocol. The frozen sections were removed from −20 °C, thawed to room temperature, and then rinsed in PBS for 5 min. They were fixed with 4% PFA at RT for 10 min, followed by washing with PBS twice for 5 min each. Next, the sections were treated with 0.3% Triton X-100 (Sangon Biotech, A600198-0500, Shanghai, China) in PBS for 5 min and rinsed again with PBS twice for 5 min each. Finally, the sections were blocked at room temperature for 2 h using a blocking buffer containing 3% BSA-V (Solarbio, A8020, purity 97.00%, Beijing, China), 0.3% Triton X-100, and 2% denatured goat serum (Beyotime, C0265, Shanghai, China), incubated with the 1st antibody in blocking buffer at RT for 1 h, and finally counterstained with 1 μg/mL DAPI (Solarbio, C0060, Beijing, China). Myosin heavy chain (Bioss, bs-5885R, dilution 1:200, Beijing, China) for Myh1 was used as the 1st antibody. Goat antirabbit IgG secondary antibody, Alexa Fluor 647 (Beyotime, A0468, polyclonal, dilution 1:500, Shanghai, China) was used as the 2nd antibody.

To visualize lipid droplets, the sections underwent fixation with 4% PFA at RT for 30 min, followed by two washes in PBS for 5 min each, incubation with 4 μM BODIPY™ 493/503 (Thermo Fisher, D2191, Suzhou, China) lipid dye at RT for 30 min, and a final counterstaining with 1 μg/mL DAPI.

### 2.12. Myogenic and Adipogenic Induction with Small Molecular Chemical Treatment

Myogenic induction: Myogenic induction was divided into two stages. In the first stage, Forskolin-treated cells were treated with DMEM/F12 medium containing 15 μM 5-Azacytidine (MCE, HY-10586, purity 99.91%, Shanghai, China), 10 μM Forskolin, 10 nM LY411575 (MCE, HY-50752, purity 98.68%, Shanghai, China), 5 μM RepSox (Targetmol, T6337, purity 99.62%, Shanghai, China), and 2 μM CHIR99021 (Targetmol, T2310, purity 99.29%, Shanghai, China) for 2 days. Subsequently, F12 medium (Biosharp, BL311A, Hefei, China) containing 2% fetal bovine serum was used for induction for 2 days in an incubator containing 5% CO_2_ at 27 °C to complete differentiation.

Adipogenic induction: Forskolin-treated cells were treated with DMEM/F12 medium containing 10% HS (horse serum, Biosharp, BL209A, Hefei, China), 10 μg/mL insulin (Targetmol, I189675, ≥27 USP units/mg, Shanghai, China), 0.5 μM IBMX (Targetmol, T1713, purity 99.86%, Shanghai, China), 0.25 μM dexamethasone (Targetmol, T0947L, purity 99.88%, Shanghai, China), and 1% Lipid Mixture (Peprotech, LM-200, Cranbury, NJ, USA) for 3 days in an incubator containing 5% CO_2_ at 27 °C.

### 2.13. Immunofluorescent and Histological Staining of Myogenic and Adipogenic Induction Cells

Immunofluorescence and Histological staining protocols refer to our previous work [18]. Immunostaining was performed by a general protocol. Cells were fixed with 4% PFA (paraformaldehyde) at RT for 10 min, permeabilized with 0.3% Triton X-100 in PBS at RT for 10 min, blocked in blocking buffer for 60 min, incubated with the 1st antibody in blocking buffer at RT for 2 h or at 4 °C overnight, incubated with the 1st antibody in blocking buffer at RT for 1 h, and finally counterstained with 1 μg/mL DAPI. Desmin (Bioss, bs-1026R, dilution 1:200, Beijing, China) for myofiber/myotube was used as the 1st antibody. Goat antirabbit IgG secondary antibody, Alexa Fluor 488 (Beyotime, A0423, polyclonal, dilution 1:500, Shanghai, China) wasused as the 2nd antibody.

To observe lipid droplets in adipocytes, the cells were fixed with 4% PFA at RT for 30 min, washed twice with PBS (5 min each), incubated in 60% isopropanol for 2 min, and then stained with Oil Red O (Aladdin, O104972, Shanghai, China).

## 3. Results

### 3.1. Primary Culture and Subculture of TCCF2022 Cells

On the sixth day of primary culture, the cells began to migrate out of the tissue mass and formed monolayers after a continued culture for a further five days. During this process, fibroblast-like and epithelioid cell morphology can be observed. The primary cells were subcultured in a ratio of 1:3, and the morphology of the cells was mainly fibroblast-like cells. It is worth noting that as the number of passages increased, cell morphology also changed, with some cells becoming smaller but remaining spindle-shaped. However, by the 20th generation, most cells had become triangular or short spindle-shaped (see Figure 1a). This suggests that the function and characteristics of the cells may change with the increase in the number of passages.

### 3.2. Characterization of TCCF2022 Cell Line

To characterize the TCCF2022 cell line, the CCK-8 assay was first used to assess the 20th-generation TCCF2022 cells’ proliferative properties, and the proliferation curve was plotted (Figure 1b). The results showed that TCCF2022 cells have high proliferative activity and can rapidly grow *in vitro*. The doubling time was approximately 25.2 h. Karyotype analysis revealed a chromosome number of 2n = 48 (Figure 1c), consistent with previously reported results [28]. Notably, TCCF2022 cells also exhibited a unique phenomenon during cultivation. Despite 100% confluency in the culture, their proliferation did not cease. Some cells began to aggregate and form colonies, resembling the proliferation of pluripotent stem cells (Figure 2a) [29]. This phenomenon suggests that TCCF2022 cells may possess characteristics similar to pluripotent stem cells. Therefore, alkaline phosphatase staining was employed to preliminarily assess their pluripotency. The results showed weak alkaline phosphatase expression (weak red staining) when the cells were grown as a monolayer on day 1. As time progressed (day 3), cells began to accumulate locally and form multilayered structures, leading to a gradual increase in alkaline phosphatase expression. By day 5, the areas of cell accumulation exhibited clear alkaline phosphatase positivity, and by day 7, strong alkaline phosphatase expression was detected in the cell aggregates and their surroundings (Figure 2b). This compelling observation strongly implies the presence of robust pluripotency within TCCF2022 cells. Then, RT-PCR was used to detect the expression of pluripotency marker genes, like *Sox2*, *Nanog*, *Oct4*, and *Klf4*, in TCCF2022 cells after 7 days of culture. As expected, these pluripotency marker genes were expressed to varying degrees.

### 3.3. Recovery of Pluripotency and Proliferation of TCCF2022 Cells

As the passage number increases, the spontaneous aggregation phenomenon of TCCF2022 cells gradually weakens and eventually disappears, and occasional instances of post-passage cells being unable to proliferate are observed. This suggests that the characteristics of the cells may have changed during the culture process. Therefore, a karyotypic analysis of TCCF2022 was performed, revealing a diploid chromosome number of 2n = 48, in agreement with prior studies, which confirms genomic stability. Thus, we speculate that TCCF2022 may undergo differentiation during the culture process. Interestingly, in our attempt to induce myogenesis in TCCF2022 using the small-molecule compound combination VCTFR (Valproic acid, CHIR99021, Tranylcypromine, Forskolin and RepSox) [30], it was observed that when individual factors were applied to induce differentiation, TCCF2022 cells regained the ability to form aggregates by day 3 post-treatment with 5 μM Forskolin, accompanied by positive alkaline phosphatase staining (Figure 3a). The relative expression levels of pluripotency marker genes were subsequently analyzed by qRT-PCR, revealing that a significantly elevated expression of *Sox2*, *Oct4*, *Klf4*, and *Nanog* was detected in the Forskolin-treated group compared to the control group (Figure 3b). Subsequently, the concentration of Forskolin was adjusted, and it was found that TCCF2022 cells exhibited favorable growth when treated with 0.5 μM Forskolin. After a long-term culture, by the 20th passage, Forskolin-treated TCCF2022 cells retained normal proliferation capability, whereas TCCF2022 cells without Forskolin treatment lost their proliferative ability (Figure 3c). These results indicate that Forskolin may promote the restoration of pluripotency in TCCF2022 cells that have been passaged for multiple generations.

### 3.4. Transcriptomics Analysis of Pluripotency Maintenance of TCCF2022

To elucidate the molecular and cellular mechanisms through which Forskolin affects the pluripotency of TCCF2022 cells, we conducted transcriptome analysis using RNA-seq on cells treated with Forskolin (Figure 4). From the volcano plot (Figure 4b), it can be observed that the expression levels of c-Myc, and Klf4 in TCCF2022 cells significantly increased after Forskolin treatment, while the expression levels of other pluripotency-related genes could not be displayed at corresponding levels due to the incomplete annotation of the spotted gar genome. GO analysis revealed the significant enrichment of GO terms related to growth, development, cell migration, differentiation, and regeneration (Figure 4c). By categorizing the differentially expressed genes into upregulated and downregulated gene sets and performing KEGG pathway enrichment analysis based on these two gene sets, it was found that several signaling pathways were significantly altered. The KEGG enrichment analysis results for upregulated genes (Figure 4d) showed significant changes in the cAMP, TGF-beta, TNF, MAPK, Wnt, JAK-STAT, FoxO, Hippo, and NF-kappa B signaling pathways. On the other hand, the KEGG enrichment analysis results for downregulated genes (Figure 4e) indicated significant changes in the PI3K-Akt, Notch, MAPK, Rap1, cGMP-PKG, and TGF-beta signaling pathways. Notably, the TGF-beta and MAPK signaling pathways were significantly altered in both the upregulated and downregulated gene sets in the KEGG analysis results. Based on the roles of genes in the signaling pathways, it was observed that the MAPK signaling pathway was overall activated, while the TGF-beta signaling pathway could be divided into the TGF-beta and BMP parts, with the TGF-beta signaling pathway being inhibited and the BMP signaling pathway being activated. Furthermore, cAMP signaling was highly activated, demonstrating the conservative role of Forskolin in activating adenylate cyclase in Topmouth culter (Culter alburnus).

### 3.5. Myogenic and Adipogenic Potential of TCCF2022

To assess the potential of caudal fin cells as seed cells for cell-cultured meat, Forskolin-treated TCCF2022 cells were further cultured. By day 15, aggregated cell colonies detached from the dish surface and formed 3D spheroids (Figure 5a). The HE staining of these 3D spheroids revealed significant morphological diversity among internal cells (Figure 5b). Specifically, distinct cell morphologies were observed myotube-like cells, adipocyte-like cells, epithelial-like cells, and neuron-like cells (Figure 5(b1–b4)), leading us to hypothesize cellular heterogeneity within the 3D spheroids. To confirm our hypothesis, immunofluorescence and histological analyses were performed. The results showed a subset of cells expressing *Myh1* (a myogenic marker) and another subset of cells containing lipid droplets (indicative of adipogenesis). These results confirm the presence of both myogenic and adipogenic lineages within the 3D spheroids, demonstrating that caudal fin cells possess differentiation potential into muscle and adipocyte.

### 3.6. Myogenic and Adipogenic Induction of TCCF2022

The spontaneous formation of myogenic and adipogenic lineage cells within the 3D spheroids demonstrated limited efficiency, with additional undesired cell lineages observed, thereby failing to meet the requirements for cell-cultured meat. To address this limitation, a shift to a 2D monolayer culture system was implemented for myogenic and adipogenic induction, and differentiation protocols were optimized. Firstly, guided by reports that 5-azacytidine (5-Aza) promotes myogenic differentiation in mesenchymal stem cells [31], Forskolin-treated TCCF2022 cells were induced with 5 μM 5-Aza for 3 days, the cell growth slowed down or even stagnated, and a small number of cells differentiated into elongated myotube-like cells. Immunofluorescence staining showed no Desmin-positive signals or multinucleated myotube formation, and RT-qPCR analysis revealed the upregulated expression of the early myogenic regulator *Myf5* but no significant increase in myosin heavy chain (*Myha*) expression. Subsequently, in addition to 5-Aza treatment, we independently administered each myogenic small-molecule compound—Forskolin, LY411575, RepSox, and CHIR99021—in parallel for 3 days, followed by low-serum differentiation. The RT-qPCR analysis revealed that this combinatorial treatment enhanced the expression of *Myf5* (Figure 6a) during the induction phase and upregulated *Myha* (Figure 6b) post-differentiation. Finally, upon combining all small-molecule compounds and implementing low-serum differentiation, immunofluorescence staining confirmed Desmin-positive multinucleated myotubes (Figure 6c), demonstrating successful terminal myogenic differentiation.

For adipogenic induction, Forskolin-treated TCCF2022 cells treated with an adipogenic cocktail exhibited rapid lipid droplet accumulation within 3 days (Figure 6d).

These results demonstrate that TCCF2022 cells can be efficiently directed to differentiate into functional myotubes and adipocytes under optimized conditions, establishing caudal fin-derived cells as a promising novel seed cell source for cell-cultured meat applications.

## 4. Discussion

Cellular agriculture technology is gradually becoming an innovative means to obtain high-quality alternative proteins, offering a sustainable solution for global population growth. Fish has always been an important source of animal protein, meeting people’s daily nutritional needs. However, traditional fisheries face challenges related to environmental and resource constraints. Therefore, the advancement of cell-cultured meat as an emerging industry, driven by stem cell technology, presents enormous potential. Stable, continuously expandable seed cells form the foundational cornerstone for scaling cell-cultured meat production. Edible meat in daily life is usually composed of muscle tissue, fat tissue, and connective tissue. Cell agriculture in mammals [5] and birds [7] usually uses MuSCs as seed cells. In fish, since fish fat contains long-chain omega-3 polyunsaturated fatty acids and can provide special flavor to fish, in order to improve the taste of cell-cultured meat and provide higher nutritional value, AD-MSCs and MuSCs are often used as seed cells for cell aquaculture. However, fish cell culture technologies remain underdeveloped, with stable, continuously passaged cell lines of MuSCs and AD-MSCs being difficult to establish across many fish species. The scarcity of reliable seed cell sources persists as a critical bottleneck. Notably, threadsail filefish (*Stephanolepis cirrhifer*) caudal fin cells have been reported to generate myotubes and adipocytes *in vitro* [25], suggesting their potential as novel seed cells for cell-cultured meat that could circumvent species-specific limitations in conventional cell line development. Moreover, using fin tissue to obtain seed cells eliminates the need to sacrifice fish, thereby supporting biodiversity conservation and addressing potential ethical concerns associated with traditional methods.

Here, we report for the first time the establishment and characterization of the caudal fin cell line TCCF2022 from Topmouth culter (*Culter alburnus*). In primary culture, epithelial-like and fibroblast-like cells migrated out from tissue blocks, with fibroblast-like cells gradually becoming dominant as passages progressed. These cells proliferated on average every 2–3 days at 27 °C, spontaneously aggregating to form 3D spheroids. Alkaline phosphatase staining showed positivity, and the cells exhibited the expression of pluripotency marker genes, including *Sox2*, *Nanog*, *Oct4*, and *Klf4*. This suggests that TCCF2022 cells possess strong pluripotency. Since the genome of Topmouth culter has not been well assembled and annotated, we first mined and annotated the unannotated genes *Sox2*, *Nanog*, *Oct4*, *Klf4*, and *Myha* from its genome sequence and SRA database (see Appendix A). However, with the increase in the number of transmission passages, the proliferation ability of the cells gradually decreased, up to about the 30th passage, when the cells even stopped growing. In an attempt to restore the cells’ ability to proliferate, we unexpectedly found that Forskolin as a single factor contributed to the recovery and maintenance of pluripotency, leading to the re-emergence of 3D growth in multi-passaged TCCF2022 cells. Other pluripotency features, such as *Sox2*, *Oct4*, *Nanog*, and *Klf4* gene expressions, significantly increased.

Forskolin, a natural compound from plants, is an activator of intracellular adenylate cyclase, increasing intracellular cyclic adenosine monophosphate (cAMP) levels [32], triggering a series of biochemical reactions, such as protein kinase A (PKA), influencing cell function and behavior, and is commonly used as one of the components for fibroblast reprogramming [33,34,35]. Transcriptomic analysis revealed that the cAMP signal was indeed highly activated in Forskolin-treated TCCF2022 cells. However, whether Forskolin-treated TCCF2022 cells can be used as a seed cell source for artificial meat needs to be further determined.

We observed the emergence of myogenic and adipogenic lineage cells within the 3D spheroids, suggesting that Forskolin-treated TCCF2022 cells have the potential to differentiate into myotubes and adipocytes. But due to low efficiency and undesired cell populations with undefined identities, it is not currently possible to apply 3D organoid-like culture directly to the formation of cell-cultured fish meat. Therefore, we strategically directed our focus to first establishing the ways of transdifferentiation of caudal fin cells into myotubes and adipocytes in a 2D monolayer system. Although the differentiation of caudal fin cells into myogenic lineage cells and adipogenic lineage cells in a 2D monolayer system has already been reported, the proprietary composition of their myogenic induction medium is not clear or optimized for fish species, and the resulting cell identities also lack characterization [25]. Moreover, the relevant induction efficiencies were not explicitly presented. Therefore, we attempted to develop a cost-effective, small-molecule cocktail with well-defined mechanisms, which could be beneficial to the cell-cultured fish meat research field.

In our optimized small-molecule cocktail (5LRCF), 5-Aza promotes myogenic differentiation by covalently binding to DNA methyltransferases (DNMTs), leading to the demethylation of the promoter regions of key myogenic genes (such as MyoD, Myogenin, and Myf5), which releases gene silencing and activates myogenic regulatory factors [31,36,37]. However, our data reveals that treatment with 5-Aza alone only marginally enhances the myogenic capacity of Forskolin-treated TCCF2022 cells. Thus, we combined 5-Aza with complementary myogenic enhancers such as Forskolin, LY411575, RepSox, and CHIR99021. Forskolin was applied at optimized concentrations to enhance pluripotency-related gene expression, whereas LY411575 and RepSox have the ability to enhance the myogenic differentiation efficiency of MuSCs in large yellow croaker (*Larimichthys crocea*) by the inhibition of Notch and TGF-β signaling, respectively [16]. LY411575 inhibits Notch signaling, thereby mimicking the natural downregulation of this pathway that occurs when muscle stem cells transition from a stem cell fate to a differentiation state. RepSox inhibits the TGF-β signaling pathway, preventing the differentiation of muscle stem cells into myofibroblasts. The purpose of the addition of CHIR99021 is to activate the Wnt/β-catenin signaling pathway, upregulating the key membrane fusion proteins Myomaker and Myomerger, which drive myocyte membrane fusion, thereby promoting myotube formation [38]. Finally, low-serum differentiation further promoted the formation of myotubes. Our combinatorial approaches successfully generated Desmin-positive multinucleated myotubes and lipid-laden adipocytes, enhancing differentiation efficiency remains a core focus of our current research. However, the molecular basis and mechanisms of Forskolin-treatment remain to be further investigated.

Although scaling up the production of cell-cultured fish meat using TCCF2022 cells still faces substantial challenges, including the development of serum-free culture media, the optimization of cellular immortalization strategies, the construction of suspension culture systems, and the determination of the optimal bioreactor configurations, the TCCF2022 cell line was successfully established from the caudal fin of Topmouth culter and demonstrated robust proliferative capacity and multipotent differentiation potential, endowing it with significant value as seed cells for cell-cultured fish meat production. Furthermore, this method is effective and has robust manipulability to maintain its pluripotency and myogenic/adipogenic induction, offering another option for producing cell-cultured meat from economically important fish species, where MuSCs and AD-MSCs lines are difficult to established.

## 5. Patents

We have applied for the relevant intellectual property protection of the methods used for establishing the cell lines and the protocols used for inducing differentiation described in this manuscript.

## Figures and Tables

**Figure 1 foods-14-02075-f001:**
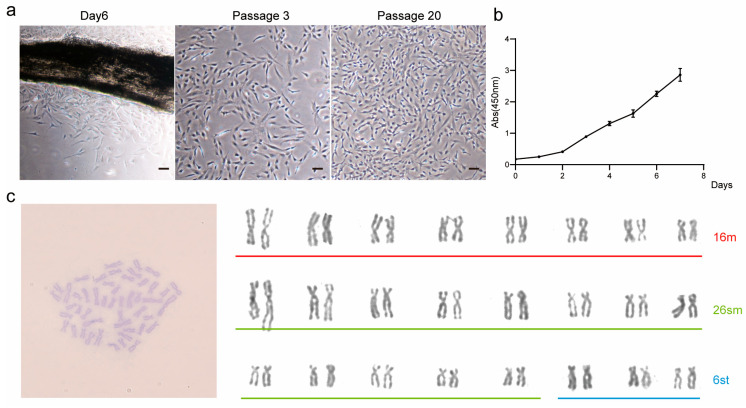
Culture of TCCF2022. (**a**) Cell morphology on the 6th day of primary culture, the 3rd generation, and the 20th generation. Scale bars: 80 µm. (**b**) The 20th-generation cell growth curve was drawn using the CCK-8 method. (**c**) Karyotype analysis of 10th-generation TCCF2022 cells. Red represents metacentric chromosomes (m), green represents submetacentric chromosomes (sm), and blue represents subtelocentric chromosomes (st).

**Figure 2 foods-14-02075-f002:**
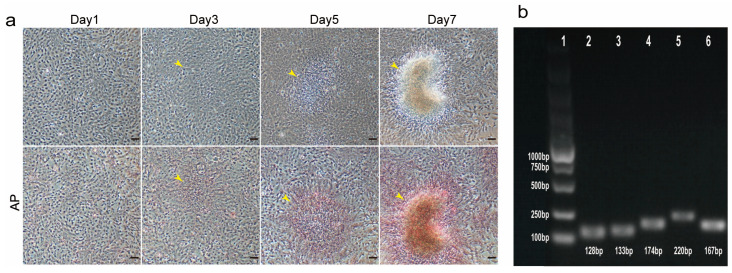
Detection pluripotency of TCCF2022. (**a**) Bright field and AP staining images of 10th-generation TCCF2022 cells at day 1, 3, 5, and 7 after they grew to monolayer; the yellow arrows represent t cell aggregation. Scale bars: 80 µm. (**b**) Electrophoretic identification of 10th-generation TCCF2022 cells by RT-PCR. 1: marker; 2: *Sox2*; 3: Nanog; 4: *Oct4*; 5: *c-Myc*; 6: *Klf4*.

**Figure 3 foods-14-02075-f003:**
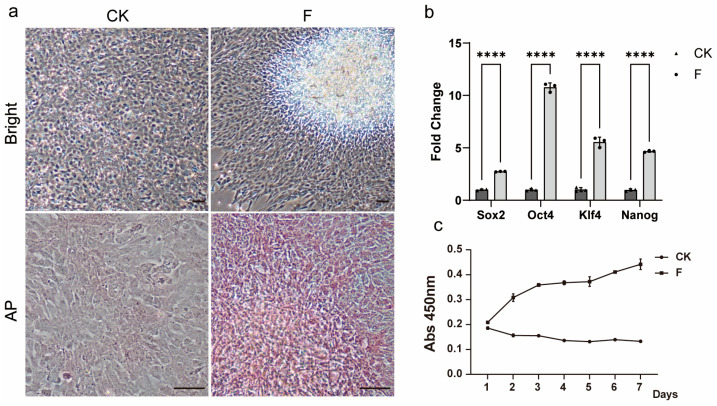
Forskolin induction can improve the pluripotency of TCCF2022. (**a**) The 20th-generation TCCF2022 cells were treated with 5 μM Forskolin for 3 days, and the results of bright field and AP staining are shown in the control group and the experimental group. Scale bars: 80 µm. (**b**) The 20th-generation TCCF2022 cells were treated with 5 μM Forskolin for 3 days, and the relative expression of stem cell marker gene in the experimental group and the control group was detected by qRT-PCR. Statistical analysis was performed on relevant data using two-way ANOVA. Error bars indicate s.d.; **** *p* < 0.00005. (**c**) The 20th-generation TCCF2022 cells were treated with 5 μM Forskolin for 7 days, and a growth curve was drawn, with circles representing the treatment group and triangles representing the control group.

**Figure 4 foods-14-02075-f004:**
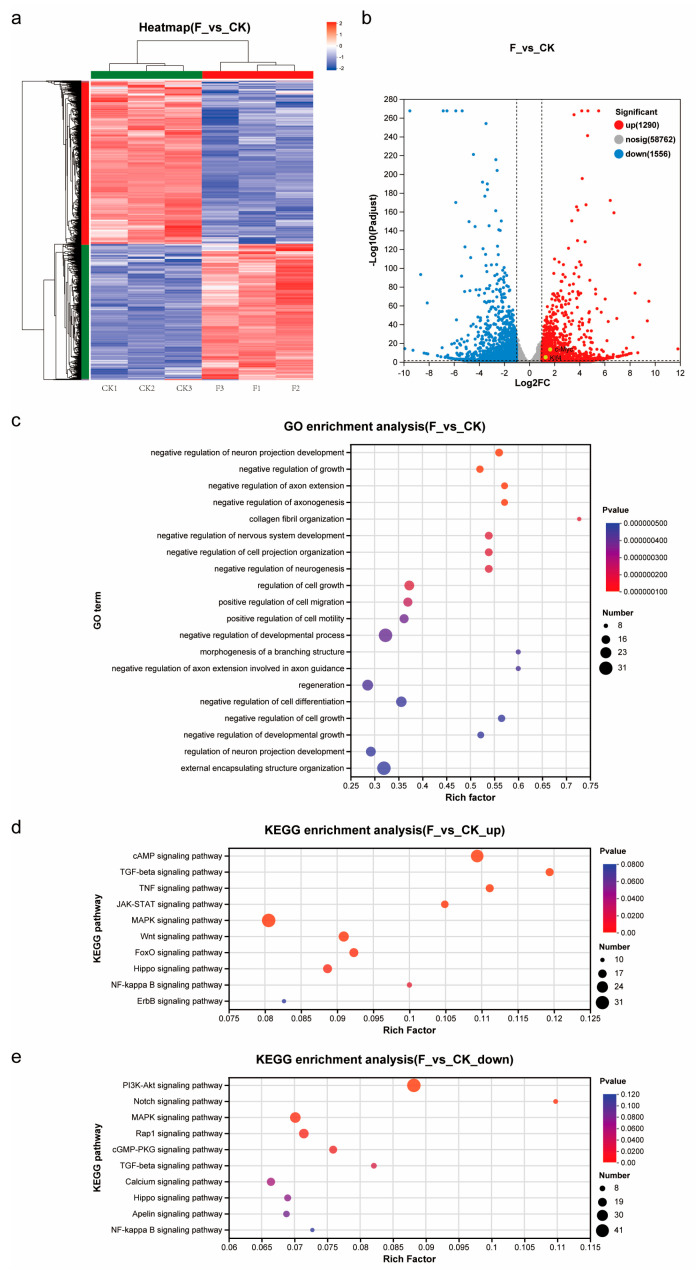
Transcriptomics analysis. (**a**) Heatmap showing the z value of all differentially expressed genes between CK and F (5 μM Forskolin treatment for 7 days). Genes (rows) and samples (columns) were clustered using average linkage clustering with Euclidean distances. (**b**) Volcano plot showing differentially expressed genes between CK and F (5 μM Forskolin treatment for 7 days). Significantly differentially expressed stem cell marker genes are highlighted. (**c**) GO enrichment indicating gene ontology terms (biological processes) corresponding to genes differentially expressed between CK and F (5 μM Forskolin treatment for 7 days). (**d**) KEGG enrichment indicating gene signal transduction pathways corresponding to upregulated differentially expressed genes between CK and F (5 μM Forskolin treatment for 7 days). (**e**) KEGG enrichment indicating gene signal transduction pathways corresponding to downregulated differentially expressed genes between CK and F (5 μM Forskolin treatment for 7 days).

**Figure 5 foods-14-02075-f005:**
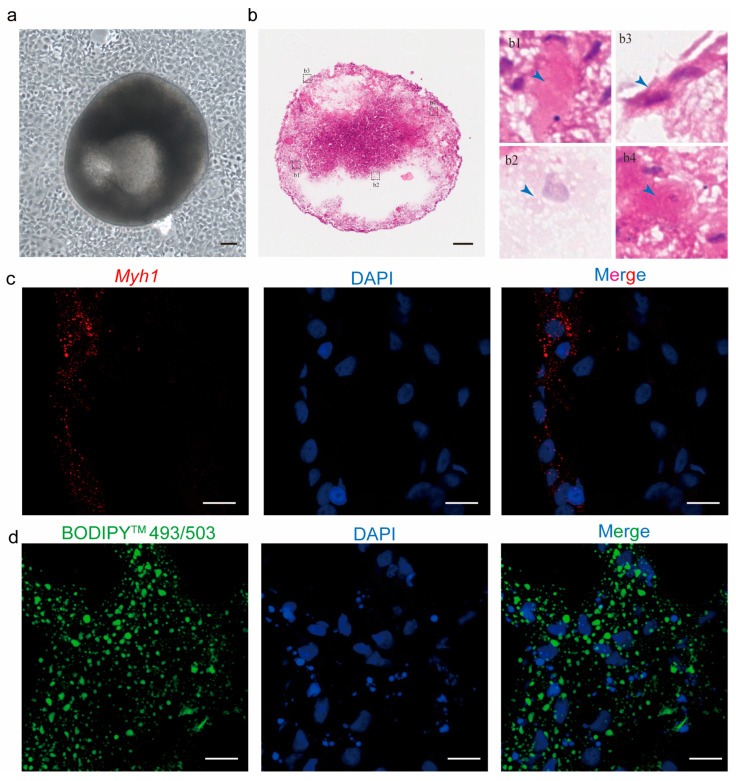
Characterization of TCCF2022-derived 3D spheroids through histology and immunofluorescence. (**a**) Spontaneous embryoid formation in Forskolin-treated cells after 7 days of culture. Scale bars: 80 µm. (**b**) HE staining of caudal fin-derived cell colonies. The outlined areas (**b1**–**b4**) from the HE staining images are enlarged in the right panels, with blue arrows showing different cell types, which are muscle-like cells, adipose-like cells, epithelial-like cells, and neuro-like cells. Scale bars: 80 µm. (**c**) Immunofluorescence images for *Myh1* expression in 3D spheroids. Cells were stained with *Myh1* (red); nuclei were counterstained with DAPI (blue). Scale bar: 10 μm. (**d**) Histological analyses of lipids in 3D spheroids. Lipids were stained with BODIPYTM 493/503 (green); nuclei were counterstained with DAPI (blue). Scale bar: 10 μm.

**Figure 6 foods-14-02075-f006:**
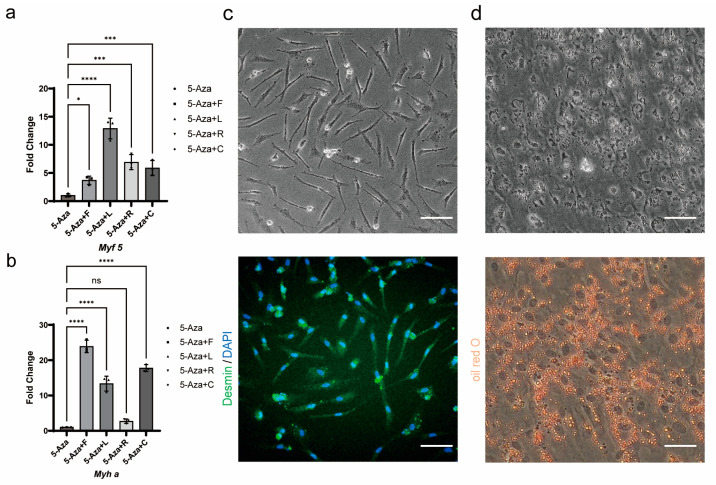
Myogenic and adipogenic induction of TCCF2022 cells. (**a**) The relative expression of Myf5 in TCCF2022 after the addition of Forskolin, LY411575, RepSox, and CHIR99021, respectively, on the basis of 5-Aza treatment was detected by qRT-PCR. (**b**) The relative expression of Myha in TCCF2022 cells pre-treated with a small-molecule compound after low-serum induced differentiation was detected by qRT-PCR. Statistical analysis was performed on relevant data using one-way ANOVA. Error bars indicate s.d.; ns *p* ≥ 0.05; * *p* < 0.05; *** *p* < 0.0005; **** *p* < 0.00005. (**c**) Forskolin-treated cells were treated with 15 μM 5-Azacytidin, 10μM Forskolin, 10 nM LY411575, 5 μM RepSox, and 2 μM CHIR99021 for 2 days, and F12 medium containing 2% FBS for 2 days, and the results of bright field and immunofluorescence are shown in the experimental group. Scale bars: 80 µm. (**d**) Forskolin-treated cells were treated with DMEM/F12 medium containing 8% HS for 3 days, and the results of bright field and Oil red O staining are shown in the experimental group. Scale bars: 80 µm.

## Data Availability

The original data presented in the study are openly available in GEO, GSE289325 at https://www.ncbi.nlm.nih.gov/geo/query/acc.cgi?acc=GSE289325 (accessed on 9 June 2025). The gene sequences have been uploaded to NCBI, and the GenBank accession numbers are as follows: *Klf4* (PP259336), *Myf5* (PP259337), *Nanog* (PP259338), *Oct4* (PP259340), and *Sox2* (PP259341); the RNA-seq data has been deposited to the GEO (accession number: GSE289325).

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
