# Peer review of "Fin Cells as a Promising Seed Cell Source for Sustainable Fish Meat Cultivation"

_foods, 2025, doi:10.3390/foods14122075_

Round 1

Reviewer 1 Report

Comments and Suggestions for Authors

General Comments:

Clarity and Detail in Methodology and Discussion:

The authors should revise the methodology section by providing detailed protocols for each experiment, including the exact number of replicates, treatment conditions, and equipment used. This will enhance reproducibility. Additionally, in the discussion, the authors should expand on the implications of their findings and provide a more critical evaluation of their results relative to prior work.

Experimental Design, Data Interpretation, and Statistical Analysis:

It’s essential to include detailed descriptions of the statistical methods used. For example, the number of replicates, type of statistical tests (e.g., ANOVA, t-test), and criteria for significance should be clearly defined. This will improve the scientific rigor and allow readers to better assess the reliability of the findings.

Novelty and Originality:

The authors should explicitly clarify how their study adds new insights to the field, especially in comparison to existing literature on cell lines derived from fish species. An in-depth comparison with past work will make the novelty of their approach clearer. Consider providing a clearer research gap statement in the introduction and discussing the unique features of the TCCF2022 cell line.

Specific Comments:

Replication and Statistical Analysis:

Add the exact number of replicates used for each experimental condition.

Specify the statistical methods used (e.g., ANOVA, t-test), including the criteria for significance (e.g., p-values).

Scientific Clarity and Rationale:

Why Topmouth Culter was chosen as the source species?

Expand on the rationale for selecting Culter alburnus. What makes this species suitable for cultured fish meat? Mention its commercial value, growth rate, market demand, or unique traits that justify its selection.

Limitations of MuSCs and ADSCs in Fish and TCCF2022’s Advantages:

Provide a more detailed explanation of the challenges in using MuSCs and ADSCs for fish species, such as limited proliferation, differentiation potential, or difficulty in maintaining pluripotency. Then, explain how TCCF2022 overcomes these limitations (e.g., better proliferation, pluripotency maintenance).

Title and Novelty:

Re-evaluate the use of “novel” in the title.

While the study presents valuable work, the term “novel” may require stronger justification. Clarify the unique contributions of the study, particularly how it differs from previous research on cell lines derived from fish species. If the novelty is not substantial, revise the title to avoid overstating claims.

Abstract Suggestions:

Revise the sentence: “...supporting future studies on meat quality, nutrition, and flavor enhancement.” Specify how TCCF2022 will contribute to meat quality, nutritional value, and flavor enhancement. For example: “This cell line can be used to improve muscle fiber formation and lipid composition, potentially enhancing the nutritional profile and flavor of cultured fish meat.”

Ensure consistency in the formatting of pluripotency markers (e.g., Oct4 vs. OCT4) and chemical names (e.g., Repsox).

Introduction:

The introduction should clearly establish the research gap and rationale for using TCCF2022. Focus on how this work fills an existing gap in fish cell lines for cultured meat, particularly highlighting the limitations of previous methods and the advantages of this new approach.

Comparison with Previous Differentiation Protocols:

Include a section comparing 5LRCF cocktail with other differentiation protocols (e.g., efficiency, scalability, simplicity). This comparison will highlight the advantages and innovations of your approach.

Materials and Methods:

Provide a more detailed methodology for each experimental procedure to ensure reproducibility. This should include:

Sources and purity of chemicals and reagents.

Specific treatment conditions (e.g., duration, temperature, concentration).

Statistical analysis methods used (e.g., ANOVA, post-hoc tests).

Add citations for established protocols where applicable, so readers can verify and reproduce the methods.

Experimental Methodology:

Confirmation of Myotube and Adipocyte Differentiation:

Specify the markers or assays used to confirm the differentiation into myotubes (e.g., MyoD, MyHC) and adipocytes (e.g., PPARγ, Oil Red O).

Functional Comparison with Native Fish Tissue:

Strengthen the results by adding data comparing the functional properties (e.g., lipid composition, protein content) of differentiated cells with native fish muscle and fat. This could include biochemical or histological analysis.

Results and Discussion:

Provide deeper interpretation of the results, with a stronger focus on scientific principles. Compare your findings with recent studies in the field to place them in context.

Scalability for Industrial Application:

Discuss the scalability of the TCCF2022 platform for large-scale cultured fish meat production. Mention potential challenges (e.g., bioreactor compatibility, cost-efficiency, regulatory hurdles) and how your approach could be adapted for industrial use.

Conclusion:

Clarify the innovative aspects of the study and its potential applications.

Broader Impact: Emphasize the significance of your findings in relation to the overall goals of cultured meat production, particularly in terms of improving fish meat quality, nutritional value, and sustainability.

Reviewer 2 Report

Comments and Suggestions for Authors

The manuscript from Zongyun et al titled as “Fin Cells as a Novel Seed Cell Source for Sustainable Fish Meat Cultivation” explains the isolation of TCCF2022, a novel caudal fin-derived cell line from Topmouth culter (Culter alburnus), which expresses pluripotency markers and aggregated growth to form 3D spheroid. Forskolin supplementation enhanced pluripotency maintenance. The presence of adipogenic and myogenic lineage cells within the 3D spheroids was confirmed, demonstrating their potential as seed cells for cell-cultured meat. Using a small-molecule cocktail 5LRCF, authors differentiated TCCF2022. Authors presented the results well. However, the minor concerns are below:

  • The current title, “Fin Cells as a Novel Seed Cell Source for Sustainable Fish Meat Cultivation,” gives meaning that the manuscript has detailed information on isolation of fin cells, and application of these novel cells in fish meat cultivation, including their effectiveness in sustaining production. However, the manuscript focuses on the generation of cell lines from fin cells and their differentiation potential, without presenting experimental evidence on actual fish meat cultivation or sustainability outcomes. Therefore, considering the results provided, the authors need to revise the title to better reflect the findings of the study.
  • Abstract…”adipose-derived mesenchymal stem cells (ADSCs)”…Please check and correct as Adipose-derived mesenchymal stem cells (AD-MSCs), or adipose-derived stem cells (ADSCs)
  • Line 280, ‘This section may be divided by subheadings. It should provide a concise and precise….” It seems authors copied this paragraph from template and forgot to remove it.  
  • 2b. Provide the size distribution of marker, and genes.
  • Line 335, provide full form of VCTFR
  • Line 342, there is no description provided about the results of figure.3C. Authors cited both Fib 3B and 3C for RT-PCR results.
  • Authors did not cite figure 6 in the results sections.
  • Line 451, “Authors should discuss the results and how they can be interpreted from the perspective of previous studies and of the working hypotheses. The findings and their implications should be discussed in the broadest context possible. Future research directions may also be highlighted”. Seems copied from the template, forgot to remove it.
  • 486, supplementary fig. 1 is missing
  • Line 532, all supplementary materials are missing.

Reviewer 3 Report

Comments and Suggestions for Authors

The experimental parts of the article were designed very correctly. I have practically nothing to point out as to the content and correctness of the scientific approach. The article contains, in my opinion, some spelling errors that I have highlighted with comments in the pdf version that I attach to these notes.
The most important note,, for me, is the fact of speaking of "animal meats" when for the moment only animals give us what we universally call meat. It is equally evident that with the development of the production of cultivated meats, it will be necessary to ask ourselves at a scientific and then legislative level whether the term "meat" can also be extended to this type of product or whether it should be reserved exclusively for meat produced by animals.
My compliments to the Authors.

Comments on the Quality of English Language

The article contains, in my opinion, some writing errors that I have highlighted with comments in the pdf version that I attach to these notes.

Round 2

Reviewer 1 Report

Comments and Suggestions for Authors

The amended manuscript should be accepted.